# Cochlear Implant Evolving Indications: Our Outcomes in Adult Patients

Andrea Achena [1], Francesco Achena [2], Alberto Giulio Dragonetti [1], Serena Sechi [2], Andrea Walter Pili [2], Maria Cristina Locci [2], Giuseppe Turnu [2], Antonino Maniaci [3,*] and Salvatore Ferlito [3]

1    U.O.C di Otorinolaringoiatria ASST Grande Ospedale Metropolitano Niguarda, 20162 Milano, Italy
2    U.O.C. di Otorinolaringoiatria, P.O. CTO–Iglesias, Assl Carbonia-ATS Sardegna Italia, 09016 Cagliari, Italy
3    Department of Medical, Surgical Sciences and Advanced Technologies G.F. Ingrassia, 95123 Catania, Italy
*    Correspondence: antonino.maniaci@phd.unict.it; Tel.: +393204154576

**Abstract:** Background: The eligibility criteria for cochlear implantation are constantly evolving, following the continuous progress in technology, knowledge about cochlear implant (CI) fitting, and the possibility to preserve residual hearing. Appropriate attention should be given to asymmetric hearing loss (AHL) and single-side deafness (SSD) subjects. This study aimed to analyze cochlear implant indications and evaluate the longitudinal performance outcomes for patients with different kinds and degrees of sensorineural hearing loss. Methods: A total of 69 adult hearing loss CI recipients were included and divided into four subgroups according to our CI indication criteria. We performed objective and subjective measures, including speech perception analysis in silence and with background noise, comparing the outcomes obtained in the four groups. Results: After cochlear implant surgery, concerning the preimplantation daily listening condition, a significantly improved speech perception score in silence and noise was found in all four groups ($p < 0.05$ for all). Conclusion: CI could represent an efficient solution for patients with AHL and SSD classes.

**Keywords:** cochlear implant; evolving indications; asymmetric hearing loss (AHL); single-side deafness (SSD)

## 1. Introduction

Cochlear implant (CI) history is studded with conflicting opinions and continuous evolution [1,2]. The most important and revolutionary principle underlying CI is the direct stimulation of auditory nerves with an electrical equivalent of the sound signal [3].

Multichannel IC was developed and implemented in clinical practice following preliminary results presented by two independent research groups, providing significantly better speech understanding than single-channel ones [4–6]. Ever since, the eligibility criteria for cochlear implantation have regularly changed, following the continuous progress in technology, knowledge about CI fitting, and the possibility to preserve residual hearing [7,8]. Nevertheless, there is still debate about the indications. Currently, special interest is given to the category of asymmetric hearing loss (AHL) and single-sided deafness (SSD) [9]. Despite the broad availability of treatment options, these indications were traditionally treated with contralateral routing of a signal hearing aid (CROS-HA) or a bone conduction device (BCD), specifically a bone-anchored hearing aid (BAHA). In July 2019, the US Food and Drug Administration (FDA) approved a CI for this category of patients. However, in most countries, SSD or AHL patients remain untreated, even though CI is the only device capable of restoring bilateral stimulation to the auditory system.

Additionally, several studies demonstrated that CI could provide adults with SSD or AHL with better speech perception in spatially separated noise, better sound localization, a better quality of life (QoL), and a decreased severity and incidence of tinnitus [10,11]. Thus, the eligibility criteria during the years of treatment were periodically renewed to include more patients with hearing disabilities.

The current study aimed to analyze the CI hearing outcomes in patients with different types and degrees of sensorineural hearing loss at long-term follow-up.

## 2. Materials and Methods

### 2.1. Patient Features

A retrospective study was performed on 69 patients diagnosed with sensorineural hearing loss who underwent a CI between January 2007 and July 2019. Patients were enrolled into a single medical center. At baseline, all the patients were submitted to appropriate counseling, vaccines, radiological assessment by petrous bone high-resolution computed tomography (CT), brain and inner-ear magnetic resonance (MR), preoperative and postoperative audiological test, and speech therapy evaluation. Patient features and demographic data such as age, sex, side and duration of hearing loss, and manufacturer of CI were recorded (Table 1).

**Table 1.** Main demographic features.

| | Descriptive Statistics of the Study Population ($n$ = 69) | |
| --- | --- | --- |
| | $n$/M $\pm$ SD | % |
| **Age at Implantation** | 46.9 $\pm$ 18.1 | |
| **Duration of Hearing Loss (year)** | 28.7 $\pm$ 16.8 | |
| **Gender** | | |
| Male | 39/69 | 56.5 |
| Female | 30/69 | 43.5 |
| **Side** | | |
| Left | 40/69 | 58 |
| Right | 29/69 | 42 |
| **Manufacturer** | | |
| Cochlear (Sidney, Australia) | 45/69 | 65.2 |
| Medel (Austria, Vienna) | 24/69 | 34.8 |

Patients with prelingual hearing loss and follow-up less than 1 year were excluded.

Thus, the participants were divided into four groups according to the different type and degree of hearing loss, and functional results were compared.

Devices of two medical companies were used: Cochlear Limited (Sydney, Australia) and Med-El (Vienna, Austria). All the subjects were implanted by an expert otosurgeon (F.A.) and underwent the senior author's down-up bone bridge surgical approach under general anesthesia [12].

### 2.2. Cochlear Implant Eligibility Criteria in Adults

All the participants were divided into four subgroups on the basis of our Cochlear Implant indication criteria:

1.  Bilateral symmetric sensorineural hearing loss (SHL) with a pure-tone average (PTA) $\geq$ 70 dB (average Fz 500–4000 Hz). Speech audiometry in the open set with hearing aids (HA) $\leq$ 50% for stimulus intensity at 65 dB.
2.  Severe to profound hearing loss for the high frequencies while maintaining low frequencies, in the absence of benefit from HA (speech recognition $\leq$ 50%), treated by electroacoustic stimulation (EAS).
3.  Asymmetric hearing loss (AHL), which cannot be treated with other hearing aids such as the CROS/bi CROS system or bone conduction implant, with the ear with the most hearing problems to be implanted with CI, and the better contralateral ear to be supported by a hearing aid.
4.  Single-side deafness or unilateral severe-profound hearing loss (SSD), with eventually associated ipsilateral tinnitus. The treatment of patients with this type of hearing loss should be considered on a case-by-case basis.

The criteria for AHL and SSD were adopted according to Vincent et al.'s classification and were achieved if one ear met candidacy for a CI and the other ear did not (Table 2) [13].

**Table 2.** Audiological classification criteria for AHL and SSD candidate groups according to Vincent et al. [13].

| Classification |
|:---:|
| **SSD** |
| Poorer ear PTA [3] 70 dB HL |
| Better ear PTA >30 dB HL |
| Interaural threshold gap [3] 40 dB HL |
| **AHL** |
| Poorer ear PTA [3] 70dB HL |
| Better ear PTA >30 and <55 dB HL |
| Interaural threshold gap [3] 15 dB HL |
| SSD, single-side deafness; AHL, asymmetric hearing loss; PTA, pure-tone average; HL, hearing loss. |

In SSD, the poorer ear PTA was $\geq$70 dB HL and better ear PTA $\leq$ 30 dB HL with an interaural threshold gap of $\geq$40 dB HL. In AHL, the poorer ear PTA should be $\geq$70 dB HL, and the better ear PTA should be between 30 and 55 dB HL with an interaural threshold gap of $\geq$15 dB HL. In other words, participants were included if the implanted ear scored $\leq$50% for the free field speech recognition score at 65 dB in quiet.

*2.3. Baseline and Postoperative Assessment*

At baseline, all patients were submitted to appropriate counseling, vaccines, radiological assessment by petrous bone high-resolution computed tomography (CT), brain and inner-ear magnetic resonance (MR), preoperative and postoperative audiological test, and speech therapy evaluation [14]. The pure-tone audiometry (PTA) was performed to measure the hearing threshold at variable frequencies (500, 1000, 2000, and 4000 Hz) and the speech perception score in free field in quiet (SPS% q) and in noise (SPS% n). All measures were taken in the best condition, preoperatively with hearing aids (HA) when present and postoperatively with CI in the free field.

Patients' speech recognition skills were measured by bisyllabic speech audiometry test [15]. We performed a series of live voice-related speech tracings, administered under different conditions, and by means of recorded sentence lists. The speech material consisted of a list of Italian sentences spoken by a female speaker. The performance of cochlear implant users was measured with the speech level held constant at 65 dB and the background noise of the cafeteria adjusted to obtain two different signal-to-noise ratios: s/n = +20 and 0 dB. Speech comprehension tests were performed in an acoustically treated room with the speakers positioned directly in front of the patient (0° azimuth) for speech presentation and behind (180° azimuth) for noise presentation. To limit the amount of reverberation, sound-absorbing panels were applied to the ceiling of the room. The speech and noise signals were controlled by means of two CD players, amplified through Philips 9638 speakers (Philips Electronics Breitner Center, Amsterdam, the Netherlands).

A specific approach was used for group 3 (AHL) and group 4 (SSD), where we preoperatively further assessed the speech perception score in quiet and in noise in the poor ear condition alone (SPS% poor ear alone) and the everyday living condition (SPS% binaural hearing), while we postoperatively assessed the SPS for the CI alone (SPS% CI) and in a bimodal way (SPS% CI + HA) in the AHL group or in a binaural way (SPS% CI + Normal ear) in the SSD group. Testing was conducted with the participants' own HAs when appropriate; otherwise, they were fitted with a clinic HA. When testing the poor ear, the better ear was plugged. These audiological tests were repeated during device activation, usually 30 days after surgery, then after 3, 6, and 12 months, and subsequently once a year.

### 2.4. Statistical Analysis

All statistical analyses were performed using R software (version 3.6.2). The Wilcoxon signed-rank test with correction for ties was used to test the efficacy of treatment in each group. A two-way ANOVA model with repeated measures was performed to evaluate potential differences in the four subgroups. Specifically, we evaluated the interaction between groups and pre and post treatment. Since two measures available for each subject, a random slope was defined for each ID.

### 2.5. Statement of Ethics

The subjects gave their written informed consent. The study was approved by the Ethics Committee ATS Sardegna with protocol number 238/2020/CE CI2020 and conducted according to the principles expressed in the Declaration of Helsinki.

### 3. Results

A total of 71 patients were included in the analysis, of which two were implanted bilaterally (73 CI). Four patients did not have regular follow-up visits due to logistical reasons; therefore, statistical analysis consisted of 69 CI.

SHL group 1 was composed of 35 patients, of which two were implanted bilaterally (37 ears). The mean age at surgery was 50.4 (range 21–83, SD 16.9). EAS group 2 included nine patients with a mean age at the time of surgery of 64.6 (range 40–79, SD 11.3). AHL group 3 consisted of 15 patients with a mean age at surgery of 57.2 (range 34–78, SD 11.8). SSD group 4 included eight patients with a mean age at surgery of 53.3 (range 34–64, SD 9.5). Appropriate consideration was made for AHL and SSD candidacy.

The mean preoperative hearing threshold in the implanted ear (PTA at 0.5, 1, 2, and 4 kHz) was 101.4 dB in group 1, 73.5 dB in group 2, 92.5 dB in group 3, and 89.6 dB in group 4.

After the implantation, the mean postoperative pure-tone audiometry (PTA at 0.5, 1, 2, and 4 kHz) with the speech processor on or with bimodal stimulation (based on the patients' habits) was 24.4 dB in group 1, 36.1 dB in group 2, 27.2 dB in group 3, and 29.3 dB in group 4 ($p < 0.001$).

At subgroup analysis among postoperative outcomes, group 1 benefitted from greater PTA outcomes than other groups ($p < 0.001$ for all).

Concerning speech perception skills, the mean pre-operative speech perception score in quiet (SPS% q) was 9.7% in group 1 and 21.8% in group 2. The mean pre-operative speech perception score with background noise (SPS% n) was 2.8% in group 1 and 17% in group 2 (Table 3).

**Table 3.** Group 1 and group 2 mean preoperative and postoperative pure-tone audiometry threshold and speech perception score in quiet and background noise. Abbreviations: bilateral symmetric sensorineural hearing loss (SHL); severe to profound hearing loss for the high frequencies electroacoustic stimulation (EAS).

| | Preop PTA Mean | Postop PTA Mean | Preop SPS % q Mean | Postop SPS % q Mean | Preop SPS% n Mean | Postop SPS% n Mean |
|---|---|---|---|---|---|---|
| *Group 1 SHL n = 37* | 101.4 | 24.4 | 9.7 | 75.6 | 2.8 | 53.8 |
| *Group 2 EAS n = 9* | 73.5 | 36.1 | 21.8 | 71.8 | 17 | 65 |

At subgroup analysis for SPS% outcomes in group 1 and 2, no statistical differences were found ($p = 0.023$). Instead, group 3 vs. group 4 demonstrated better postoperative outcomes than the other subgroups analyzed ($p = 0.801$).

Further assessment was made for groups 3 and 4. We evaluated both the poor ear-only condition and the everyday listening condition. The mean preoperative speech perception

score in quiet in the poor ear only and the everyday living condition was 9.2% and 54.2% in group 3, respectively, and 6.2% and 63.7% in group 4, respectively. The mean preoperative speech perception score with background noise in the poor ear only and the everyday living condition was 5.7% and 49.3% in group 3, respectively, and 2.5% and 52.5% in group 4, respectively.

Regarding speech perception skills, the mean postoperative speech recognition score in quiet was 75.6% in group 1 and 71.8% in group 2. The mean postoperative speech perception score with background noise was 53.8% in group 1 and 65% in group 2 (Table 2).

After the implantation in groups 3 and 4, we assessed both the cochlear implant (CI) ear only and the bimodal (CI+HA group 3) or binaural (CI + normal ear group 4). The mean postoperative speech perception score in quiet in the cochlear implant ear only and in the bimodal/binaural cases was 40.7% and 78.5% in group 3, respectively, and 41.2% and 87.5% in group 4, respectively. The mean postoperative speech perception score with background noise in the cochlear implant ear only and in the bimodal/binaural cases was 29.3 and 71.4% in group 3, respectively, and 33.7 and 77.5% in group 4, respectively (Table 4).

**Table 4.** The group 3 asymmetric hearing loss (AHL), and group 4 single-side deafness (SSD) mean preoperative and postoperative pure tone audiometry (PTA) threshold and speech perception score in quiet (SPS % q) and with background noise (SPS % n), tested preoperatively in the poor ear only and in the everyday living condition (binaural hearing) and postoperatively in cochlear implant ear (CI) and in bimodal (CI+HA) or binaural (CI+normal ear) condition.

| Preop PTA Mean/SD | Postop PTA Mean/SD | p | Preop SPS % q Mean Poor Ear only/SD | Postop SPS % q Mean CI/SD | p | Preop SPS % q Mean Everyday Living Condition/SD | Postop SPS % q Mean Bimodal/Binaural/SD | p | Preop SPS% n Mean Poor Ear only/SD | Postop SPS% n Mean CI/SD | p | Preop SPS% n Mean Everyday Living Condition/SD | Postop SPS% n Mean Bimodal/Binaural/SD | p |
|---|---|---|---|---|---|---|---|---|---|---|---|---|---|---|
| 92.5/0.7 | 27.2/1.1 | 0.0005 | 9.2/7.3 | 40.7/8 | 0.0004 | 54.2/9.3 | 78.5/10 | 0.001 | 5.7/6.4 | 29.3/9.9 | 0.0004 | 49.3/9.1 | 71.4/7.7 | 0.001 |
| 89.6/5.1 | 29.3/2 | 0.006 | 6.2/7.4 | 41.2/8.3 | 0.006 | 63.7/8.8 | 87.5/10.3 | 0.006 | 2.5/4.6 | 33.7/10.6 | 0.006 | 52.5/7 | 77.5/8.8 | 0.006 |

## 4. Discussion

Cochlear implant candidacy has evolved tremendously since the second half of the 1980s when multichannel implants were approved for adults [16,17]. Previously, patients were required to have a total bilateral or profound bilateral sensorineural hearing loss to be considered a candidate for cochlear implantation. Following the improvements in technology, the candidacy criteria changed and now include individuals with significant levels of residual hearing. Our experience and indications for CI candidacy criteria changed over time, initially treating only patients with total or profound deafness and then expanding the indications to patients with severe bilateral HL. Consequently, as the expanding goal of CI is to maintain or preserve hearing, patients with normal hearing or moderate hearing loss in low frequency and severe to profound sensorineural hearing loss in high frequency were also treated using the electroacoustic (EAS) or hybrid stimulation, as also proposed by other authors [18].

It is essential to use an atraumatic surgical technique in these patients including a very delicate opening of the cochlea, with slow insertion of thin and sometimes shorter than usual array electrodes [19,20]. This indication is especially important for older patients since aging also reduces hearing clarity with a well-fitted hearing aid. In such patients, postoperative electroacoustic stimulation (acoustic stimulation in the low-frequency range and electric in the high-frequency range) allows a better understanding of speech in noise and the perception of music. The CI can improve speech understanding for this kind of patient. Pillsbury et al. (2018), in a multicenter prospective study on 73 patients, with residual low-frequency hearing and severe to profound hearing loss in the mid to high frequencies treated with EAS, reported that 85% of the subjects performed better in the SPS% at 12 months in the EAS condition compared with the preoperative aided condition [21]. Our results confirmed this result in the EAS group, reporting that the average percentage of SPS in silence changed from 21.8% before surgery to 71.8% after surgery, while, in noise, it changed from 17% to 65% ($p < 0.001$). Studies showed that bimodal hearing restores some binaural cues providing superior speech recognition in noise and localization of sounds [22,23]. Franko-Tobin et al. (2015) in a retrospective study on 35 patients suffering from AHL and implanting the worse hearing ear with severe to profound hearing loss reported a speech recognition score changing from 5.4% preoperatively to 40% postoperatively [24]. The outcome in our AHL group is in line with this result, whereby the mean speech perception score in the poor ear improved from preoperative 9.2% to postoperative 40.7%.

Moreover, we further expanded CI indications by including emerging indications recommended by some authors, such as patients with SSD and associated tinnitus [25,26]. We saw that CI improved hearing and reduced tinnitus in most of these patients. Consistent with Garcia et al., we achieved great results in the SPS% everyday living conditions in silence and noise [27]. The mean preoperative speech perception score in noise in the everyday hearing condition was 52.5%; postoperatively, the mean scores were 77.5%.

As stated by Firszt et al. (2018), the rise in speech recognition for post-implant everyday living conditions AHL groups might be linked to two main traits: access to high-frequency information and benefit from binaural summation effect [28]. Prior to -implant, most of the patients were unable to ear words at high frequency, fundamental for speech clarity, whereas post-surgery, they all had high frequency in at least one ear. In addition, it is well documented that sound presented to both ears is perceived as being louder than the same signal presented to a single ear [29]. This psychophysical effect is termed binaural loudness summation. Therefore, since the listening condition of most pre-surgery patients was better in one ear only, the improvements in hearing performance were associated with the effect of binaural summation, re-established after surgery.

*Study Limitations*

A major limitation in our study was the presence of an insufficiently sophisticated speech perception test. This could severely limit the inferences and conclusions that

have been reported. Standard tests of speech perception in noise may overestimate the performance of cochlear implant wearers in the real world. Moreover, data on speech intelligibility under realistic conditions might differ in real-life performance, speech, and common noise levels, demonstrating lower performance.

## 5. Conclusions

The cochlear implant candidacy criteria have evolved. Previously, patients were required to have a bilateral total or profound bilateral sensorineural hearing loss to be considered candidates for cochlear implantation, whereas we currently include individuals with significant levels of residual hearing. Following previous data, our experience with CI candidacy shows this changing trend. The subgroups that we treated over time, from SHL patients only to the more recent inclusion of EAS, AHL, and SSD patients, endorse this evolution. Our results were generally satisfactory in terms of both speech performance and QoL. By treating patients with AHL or SSD and associated tinnitus, we saw that CI could improve hearing and drastically reduce tinnitus in most of the patients. Therefore, we argue that there is a need to align and apply the cochlear implant criteria on a larger scale to allow more hearing-impaired people to benefit from this treatment and avoid the pathology linked to hearing loss, such as depression or senile dementia.

These general guidelines are derived from our own experience; for this motive, they are questionable and subject to change over time.

**Author Contributions:** Conceptualization, A.A. and A.M.; methodology, S.F.; software, A.G.D.; validation, S.S., A.W.P. and A.A.; formal analysis, M.C.L.; investigation, G.T.; resources, A.M.; data curation, F.A.; writing—original draft preparation, A.A.; writing—review and editing, F.A.; visualization, S.F.; supervision, A.A., A.G.D. and A.M.; project administration, S.F.; funding acquisition, S.F. All authors have read and agreed to the published version of the manuscript.

**Funding:** This research received no external funding.

**Institutional Review Board Statement:** The study was conducted according to the guidelines of the Declaration of Helsinki and approved by the Ethics Committee ATS Sardegna with protocol number 238/2020/CE CI2020.

**Informed Consent Statement:** Informed consent was obtained from all subjects involved in the study.

**Data Availability Statement:** Not applicable.

**Acknowledgments:** The authors thank all the staff in the sector, both in the department and in the operating room, for the professionalism and humanity shown toward the implanted patients.

**Conflicts of Interest:** The authors declare no conflict of interest.

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
