# Peer review of "Cochlear Implant Evolving Indications: Our Outcomes in Adult Patients"

_audiolres, doi:10.3390/audiolres12040042_

Round 1

Reviewer 1 Report

The manuscript reports improvements in word-identification scores for CI patients, categorized by the type and severity of hearing loss. The results, briefly, were that the patients could identify more words after receiving a CI, as everyone would expect.  The speech perception test itself was not very sophisticated (and also not described very well).   That places a severe constraint on the inferences and conclusions that might have been possible.  The Introduction and the Discussion include historical information that might be interesting in another context, but here those sections are longer than justified by the meager data.

Author Response

Reviewer 1

The manuscript reports improvements in word-identification scores for CI patients, categorized by the type and severity of hearing loss. The results, briefly, were that the patients could identify more words after receiving a CI, as everyone would expect.

Comment: The speech perception test itself was not very sophisticated (and also not described very well).   That places a severe constraint on the inferences and conclusions that might have been possible. 

Response: Dear editor, thanks for the suggestion. As considered, we’ve considered the limitations regarding the speech perception test and argumented the main issues in regard in a specific section named ‘’Study Limitations’’.

Moreover, we described better in the methods the test and cited a reference ‘’ Patients' speech recognition skills were measured by bisyllabic speech audiometry test[15]. We performed a series of live voice-related speech tracings, administered under different conditions, and by means of recorded sentence lists. The speech material consisted of a list of Italian sentences spoken by a female speaker. The performance of cochlear implant users was measured with the speech level held constant at 65 dB and the background noise of the cafeteria adjusted to obtain two different signal-to-noise ratios: s/n = +20 and 0 dB. Speech comprehension tests were performed in an acoustically treated room with the speakers positioned directly in front of the patient (0° azimuth) for speech presentation and behind (180° azimuth) for noise presentation. To limit the amount of reverberation, sound-absorbing panels were applied to the ceiling of the room. The speech and noise signals were controlled by means of two CD players, amplified through Philips 9638 speakers (Philips Electronics Breitner Center, Amsterdam, The Netherlands.) Center, Amsterdam, The Netherlands).’’

Comment: The Introduction and the Discussion include historical information that might be interesting in another context, but here those sections are longer than justified by the meager data.

Response: dear revisor, as indicated we removed all the storical data that not contributed improving the quality of the paper.

Reviewer 2 Report

This study showed that the CI indications are evolving as the advances of CI technologies. It provided evidence that CI can reduce the pure tone audiometry threshold and improve speech perception score in quiet environment and environment with background noise in patients with asymmetric hearing loss and single side deafness. This study argued that more people with hearing loss can benefit from CI if the CI criteria could expand. The study is overall solid. However, it would be great if Results section could be revised since it basically only described the values reported in Table 3 and 4. The readers may expect the Results section could provide more information.

Comments:

  • Abstract line 10: define abbreviation CI.
  • Grammar problem in Abstract: line 13-14. Should it be “This study aimed to analyze … and evaluate …”?
  • Abstract line 16: use CI instead of Cochlear Implant
  • Abstract line 18-20: please reword the Results to make it clearer.
  • Methods 2.4 Statistical Analysis. It looks like the study mainly discussed the difference between pre- and post-treatment for each group. I didn’t see the comparison among groups. Can the authors clarify it? For example, which group benefit more from CI than other groups?
  • Table 3 and 4 only have mean values. It would be great if the authors can include SD in the two tables.
  • Please add p values or n.s. in Table 3 and 4.
  • Line 170, Table II or Table 2. Please be consistent.
  • Line 219, 5,4% or 5.4%?
  • Line 220-221, was the increasing significant?
  • Line 240-243 is redundant since line 185-188 have already discussed it.

There are more typos and grammar mistakes in this manuscript that are not listed in the comments. Please fix them. Thanks!

Author Response

Reviewer 2

This study showed that the CI indications are evolving as the advances of CI technologies. It provided evidence that CI can reduce the pure tone audiometry threshold and improve speech perception score in quiet environment and environment with background noise in patients with asymmetric hearing loss and single side deafness. This study argued that more people with hearing loss can benefit from CI if the CI criteria could expand. The study is overall solid. However, it would be great if Results section could be revised since it basically only described the values reported in Table 3 and 4. The readers may expect the Results section could provide more information.

Comments:

  • Abstract line 10: define abbreviation CI.

Response: thanks revisor, I’ve included the abbreviation

  • Grammar problem in Abstract: line 13-14. Should it be “This study aimed to analyze … and evaluate …”?

Response: thanks, corrected

  • Abstract line 16: use CI instead of Cochlear Implant

Response: modified thanks

  • Abstract line 18-20: please reword the Results to make it clearer.

Response: corrected as follow: After cochlear implant surgery, concerning the preimplantation daily listening condition, a significantly improved speech perception score in silence and noise was found in all 4 groups (p <0.05 for all).

  • Methods 2.4 Statistical Analysis. It looks like the study mainly discussed the difference between pre- and post-treatment for each group. I didn’t see the comparison among groups. Can the authors clarify it? For example, which group benefit more from CI than other groups?

Response: thanks for the suggestions, as indicated we’ve added the subgroup analysis’’ At subgroups analysis among postoperative outcomes, the group I benefit greater PTA outcomes than other groups (p<0.001 for all).’’

And for SPS ‘At subgroup analysis for SPS% outcomes in among group 1 and 2, no statistical differ-ences were found (p=0.023). Instead, Group 3 vs. 4 demonstrated better postoperative outcomes than the others subgroups analyzed (p=0.801)’’.

  • Table 3 and 4 only have mean values. It would be great if the authors can include SD in the two tables.
  • Response: thanks for the suggestions, we’ve added the values required
  • Please add p values or n.s. in Table 3 and 4. thanks for the suggestions, we’ve added the values required
  • Line 170, Table II or Table 2. Please be consistent.

Response: corrected, thanks.

  • Line 219, 5,4% or 5.4%?

Response: thanks corrected

  • Line 220-221, was the increasing significant?

Response: we’ve removed the sentences due to the precedent revisions required of the historical data, thanks

  • Line 240-243 is redundant since lines 185-188 have already discussed it.

Response:

we’ve removed the sentence redundant and clarified the link with the precedent concept.

Round 2

Reviewer 1 Report

The authors have addressed my comments.